# Effects of Toxic Lithium Levels on ECG—Findings from the LiSIE Retrospective Cohort Study

**DOI:** 10.3390/jcm11195941

**Published:** 2022-10-08

**Authors:** Petra Truedson, Michael Ott, Krister Lindmark, Malin Ström, Martin Maripuu, Robert Lundqvist, Ursula Werneke

**Affiliations:** 1Sunderby Research Unit, Department of Clinical Sciences, Psychiatry, Umeå University, 90187 Umeå, Sweden; 2Department of Public Health and Clinical Medicine, Medicine, Umeå University, 90187 Umeå, Sweden; 3Department of Clinical Sciences, Karolinska Institutet, Danderyd University Hospital, 18288 Stockholm, Sweden; 4Department of Psychiatry, Sunderby Hospital, 97180 Luleå, Sweden; 5Department of Clinical Sciences, Psychiatry, Umeå University, 90187 Umeå, Sweden; 6Sunderby Research Unit, Department of Public Health and Clinical Medicine, Umeå University, 90187 Umeå, Sweden

**Keywords:** lithium, drug-related side effects and adverse reactions, toxicity, long QT syndrome, electrocardiography

## Abstract

(1) Background: Few studies have explored the impact of lithium intoxication on the heart. (2) Methods: We examined electrocardiogram (ECG) changes associated with lithium intoxication in the framework of the LiSIE (Lithium—Study into Effects and Side Effects) retrospective cohort study. We analysed ECGs before, during, and after intoxication. (3) Results: Of the 1136 patients included, 92 patients had experienced 112 episodes of lithium intoxication. For 55 episodes, there was an ECG available at the time; for 48 episodes, there was a reference ECG available before and/or after the lithium intoxication. Lithium intoxication led to a statistically significant decrease in heart rate from a mean 76 beats/min (SD 16.6) before intoxication to 73 beats/min (SD 17.1) during intoxication (*p* = 0.046). QTc correlated only weakly with lithium concentration (ρ = 0.329, *p* = 0.014). However, in 24% of lithium intoxication episodes, there were QT prolongations. In 54% of these, QTc exceeded 500 ms; patients with chronic intoxications being more affected. (4) Conclusions: Based on summary statistics, effects of lithium intoxication on HR and QTc seem mostly discrete and not clinically relevant. However, QT prolongation can carry a risk of becoming severe. Therefore, an ECG should always be taken in patients presenting with lithium intoxication.

## 1. Introduction

Lithium is a mainstay pharmacological treatment of bipolar disorder (BD), either during acute manic episodes, or as long-term maintenance treatment. In this capacity, the UK National Institute for Clinical Excellence (NICE) has endorsed lithium as a first line treatment [1] in 2014. Lithium is also used as an augmentation treatment for depression when antidepressants do not suffice. Observational and trial evidence suggest that, compared to other mood stabilisers, lithium is therapeutically superior for the prevention of suicide and of relapse into affective episodes [2,3,4,5,6].

Despite its clinical usefulness, lithium is not without problems. Lithium can affect kidney, thyroid, and parathyroid function, all of which require regular monitoring. A further problem is the narrow therapeutic window of lithium, with only a small difference between therapeutic and toxic concentrations. Depending on indications, the therapeutic lithium serum concentration (s-Li) lies between 0.4–1.2 mmol/L. A s-Li ≥ 1.5 mmol/L is seen as the cut-off point for a clinically significant intoxication [7]. At higher serum concentrations, lithium intoxications may become life-threatening. In our previous work, there was one case of intoxication with a s-Li ≥ 1.5 mmol/L per 100 patient-years of lithium treatment. About one third of intoxicated patients required intensive care [7]. 

The effects of lithium on the heart have rarely been studied. Due to its chemical proximity to sodium, lithium can be expected to affect cardiac conduction [8]. Particularly at supratherapeutic concentrations, lithium may affect cardiac conduction, leading to prolongation of the QT interval, bradycardia, and T-wave changes [9,10]. Lithium may even give rise to a Brugada type-1 pattern [11]. A retrospective study from France, analysing 128 patients with lithium intoxication admitted to intensive care (ICU) between 1992 and 2013, found cardiovascular complications in 24% of cases [12]. Acute care physicians and intensivists need to understand these potential cardiac risks to manage patients with lithium intoxication safely. 

### Aims and Hypothesis

We conducted the current study to examine the impact of toxic s-Li on the heart. Specifically, we tested the hypothesis that higher s-Li would lead to clinically relevant electrocardiogram (ECG) changes.

## 2. Materials and Methods

### 2.1. Study Design and Setting

This study was part of the LiSIE (Lithium—Study into Effects and Side Effects) retrospective cohort study based on a review of medical records. LiSIE aimed at identifying the best treatment for BD and related disorders, such as schizoaffective disorder (SZD), considering both effects and side effects. The study was conducted according to the guidelines of the Declaration of Helsinki and was approved by the Regional Ethics Review Board at Umeå University, Umeå Sweden (DNR 2010-227-31M, DNR 2011-228-32M, DNR 2014-10-32M, DNR 2018-76-32M).

Within the framework of this retrospective cohort study, we examined ECG at the time of lithium intoxication and compared these with reference ECG, taken before (ECG_PRE-INTOX_) and after (ECG_POST-INTOX_) intoxication. The study was first piloted [13] and then conducted according to the STROBE (Strengthening the Reporting of Observational Studies in Epidemiology) checklist (Appendix A). 

### 2.2. Participants

LiSIE invited all individuals in the Swedish regions of Västerbotten and Norrbotten ≥ 18 years of age, who had (a) received a diagnosis of BD (ICD F31), SZD (ICD F25) between 1997 and 2011, or (b) used lithium as a mood stabiliser between 1997 and 2011, for instance in the context of a depressive illness [7,14,15]. Participants were informed about the nature of the study in writing and provided verbal informed consent. The consent was documented in our research files, dated, and signed by the research worker who obtained the consent. In accordance with the ethics approval granted, for deceased patients, no consent was obtained. Recruitment was concluded by the end of 2012. The cohort was locked at this point; no new patients were included in the study thereafter.

### 2.3. Inclusion Criteria

We included all patients from the LiSIE cohort who (a) lived in the Swedish region of Norrbotten, (b) had experienced a documented episode of lithium intoxication with s-Li ≥1.5 mmol/L at any time between 1997 and 2017, and (c) had an ECG recorded at the time of the intoxication. 

### 2.4. Exclusion Criteria

We excluded episodes of supratherapeutic s-Li when it was clear that these were only transient and had not given rise to an intoxication. This could for instance occur when patients had taken their prescribed lithium by mistake before their blood-test, resulting in measurement of peak (highest) rather than trough (lowest) concentrations. 

### 2.5. Data Sources/Measurement 

Lithium concentrations were obtained from a central laboratory database. ECGs were extracted from the electronic case records. Where necessary, we complemented with data manually extracted from hardcopy case records. Prior to analysis, we anonymised the data. 

### 2.6. Outcome Variables

The outcome of this study was ECG changes observed in the context of a lithium intoxication. During the 21-year review period, some patients had repeated episodes of lithium intoxication, which were not temporally related. Therefore, we analysed ECG changes per episode and not per patient.

#### ECG Selection

For ECG_INTOX_, we chose the ECG taken at the time of the lithium intoxication. If several ECGs were available, we chose the ECG closest in time to the s-Li measurement. We then analysed reference ECGs available before intoxication, pre-intoxication, and/or after the intoxication, post-intoxication (ECG_PRE-INTOX_, ECG_POST-INTOX_) (Table 1). Using an ECG ruler, we analysed all ECGs regarding heart rate (HR), rhythm, PQ interval, QRS duration, QT interval, ST segment and T-wave changes, U-wave presence, and pacemaker rhythm (Table 2). QT interval was corrected for HR (QTc), using a calculator for Bazett’s formula [16]. We further explored occurrence of QT prolongation as a dichotomous yes/no variable for the QTc value and left ventricular enlargement according to Sokolow-Lyon criteria [17]. For men, we used 0.450 s as cut-off point for QT prolongation, and for women we used 0.470 s [18]. All ECGs were coded according to the Minnesota Code Manual of Electrocardiographic Findings [19]. A specialist in cardiology double-checked all ECGs with unclear pathological changes.

### 2.7. Exposure Variables

#### 2.7.1. Serum Lithium Concentrations

For ECG_INTOX_, to count as a clinically significant lithium intoxication, s-Li had to reach at least 1.5 mmol/L [7,20]. We chose the s-Li closest to the time of the ECG recording. Each episode of lithium intoxication was categorized as acute, acute on therapeutic, or chronic. We defined intoxications as acute when s-Li rose to ≥1.5 mmol/L within 24 h after ingestion of a supratherapeutic lithium dose (accidental or non-accidental overdose) in a lithium naïve patient. We defined intoxications as acute on therapeutic when s-Li rose to ≥1.5 mmol/L within 24 h after ingestion of a lithium overdose in a patient already treated with lithium at the time. As these intoxications were comparable in character, we merged them into one “acute” category. We regarded an intoxication as chronic when a recent overdose was ruled out [7]. Finally, we stratified severity of lithium intoxication into two categories: less severe with a lithium concentration ≤ 2.5 mmol/L, and severe with a lithium concentration > 2.5 mmol/L [7,21,22].

For the reference ECG, we chose the lithium concentration logged in the medical notes nearest in time to the respective ECG, in a range between 0.2 mmol/L (minimal evidence of exposure) to 1.2 mmol/L (maximal therapeutic concentration).

#### 2.7.2. Concomitant Use of Medicines with Potential QT Prolonging Effect

We recorded the use of medicines that could potentially prolong QT or induce torsades de pointes (TdP). These medicines were classified into three risk categories [23,24].

Known TdP risk: medicines known to prolong the QT interval with a known risk of TdP, even if taken as recommended.Possible TdP risk: medicines known to prolong the QT interval with a current lack of evidence for a risk of TdP if taken as recommended.Conditional TdP risk: medicines that may cause TdP if taken (a) in excessive dosage, (b) in conjunction with other medicines that may directly or indirectly contribute to QT prolongation, or (c) by patients with additional risk factors, such as congenital long QT interval, extreme bradycardia, or hypokalaemia.

#### 2.7.3. Concomitant Use of Medicines with Other Cardiac Effects

We recorded antiarrhythmics and antihypertensives as potential cofounders of ECG changes. 

#### 2.7.4. Cardiovascular Comorbidities and Risk Factors

We recorded the presence of cardiovascular comorbidities and risk factors coinciding with any of the included ECGs. Such conditions included cerebral stroke/transient ischemic attack (TIA), acute myocardial infarction/ischemic heart disease, heart failure, angina pectoris, arrhythmias, arterial hypertension, pacemaker use, hyperlipidaemia, and diabetes mellitus. 

#### 2.7.5. Potassium Concentrations

Where available, we also recorded potassium concentration (s-K) at the time around intoxication. We defined hypo- and hyperkalaemia according to the laboratory s-K reference intervals in use at the time of intoxication. From 1999 to May 2004, this was <3.5 mmol/L for hypokalaemia and >4.7 mmol/L for hyperkalaemia. From June 2004, this was <3.6 mmol/L for hypokalaemia and >4.6 mmol/L for hyperkalaemia. For the years 1997 to 1998, information for the reference intervals was not available. For these years, we used the reference intervals valid between 1999 and May 2004.

### 2.8. Bias

We checked for selection bias in the LiSIE cohort. In accordance with the ethics approval granted, we compared age, sex, maximum recorded lithium, and creatinine concentrations in anonymous form for consenting and non-consenting patients. There were no significant differences. For patients with lithium intoxication, we compared age and sex distribution between episodes with ECG and without ECG. Again, we did not find any significant differences. However, the mean lithium concentration was significantly higher in the group having available ECG (*p* = 0.003). This means that our sample was biased toward more severe lithium intoxications. 

### 2.9. Missing Data

By default, ECG_INTOX_ was available for all included cases. Reference ECGs were not available for all cases. We conducted separate subgroup-analyses for episodes with ECG_INTOX_ and ECG_PRE-INTOX_ or ECG_POST-INTOX_. 

### 2.10. Statistical Methods

We reported data at episode level. First, we conducted a descriptive analysis of all variables. For continuous variables, we reported mean and standard deviation (SD), as well as median and minimum/maximum (min/max). For comparison of continuous variables between ECG_INTOX_ and ECG_PRE-INTOX_ or ECG_POST-INTOX_, we used Wilcoxon rank test for paired data. For correlation between s-Li and HR or QTc at intoxication, we used Spearman Rank correlation (ρ) accounting for the non-normal distribution of the data. For categorical variables, we reported proportions. For comparison of categorical variables between ECG_INTOX_ and ECG_PRE-INTOX_ or ECG_POST-INTOX_, we used McNemar’s test. We also conducted a univariate analysis with logistic regression to explore which exposure variables were associated with QT prolongation at intoxication. We abstained from multivariate analysis because the sample size was too small for the number of variables to be fitted. This would have resulted in overfitting of the model. The significance level was set to 0.05 throughout. The statistical analysis was conducted with IBM SPSS Statistics (Version 27, IBM, Armonk, NY, USA).

## 3. Results

Of 1136 patients (673 females, 59%) exposed to lithium between 1997 and 2017, 92 patients had experienced 112 episodes of lithium intoxication with lithium concentrations ≥ 1.5 mmol/L. Seventeen patients had more than one episode of intoxication. ECG_INTOX_ was available for 55 episodes in 50 patients. For 48 episodes, there was a reference ECG available, 40 ECG_PRE-INTOX_ and 28 ECG_POST-INTOX_ (Figure 1). 

### 3.1. Baseline Characteristics

The 55 episodes occurred at a mean age of 55.2 years and a median of 56.0 years. Seventy-one percent of the patients were women, and 64% of episodes were chronic intoxications. The mean maximum s-Li was 2.4 mmol/L, and the median was 2.1 mmol/L. Twenty-nine percent of intoxications were severe. The highest s-Li recorded was 9.3 mmol/L in the context of an acute intoxication. In 49% of all episodes, ECG and s-Li were obtained within 90 min of each other. Fifty-one percent of patients had cardiovascular comorbidities, and 24% were receiving anti-arrhythmic medication at the time. Based on 51 episodes with available s-K, 24% had hyperkalaemia and 15% had hypokalaemia (Table 3). One patient died within one day of the lithium intoxication, but the death was considered unrelated to the intoxication.

### 3.2. ECG Changes at Lithium Intoxication

There was a wide variation of ECG changes, with QT prolongation and T-wave inversion being most common (Table 4).

### 3.3. Heart Rate and Rhythm

The mean HR at intoxication was 73 beats/min (SD 17.1) and the median 71 beats/min (min 42, max 119). The mean HR pre-intoxication was 76 beats/min (SD 16.6) and the median 78 beats/min (min 45, max 115). The mean HR post-intoxication was 78 beats/min (SD 16.1) and the median 79 beats/min (min 52, max 107). There was a significant decrease in HR during intoxication compared to ECG_PRE-INTOX_ (*p* = 0.046), but not to ECG_POST-INTOX_ (*p* = 0.614). HR on ECG_INTOX_ did not correlate with s-Li at the time of intoxication (ρ = 0.051, *p* = 0.710). Sinus rhythm was present in 48 (87%) of the episodes. In one of these, a one-beat sinus arrest occurred. Of the remaining seven episodes, four showed atrial fibrillations. In one of these, this was a new phenomenon. Two episodes recorded a nodal rhythm. In one of these, this was a new phenomenon. One episode had pacemaker rhythm. Two episodes had prolonged PQ interval. In one of these, this was a new phenomenon. 

### 3.4. QT Interval

The mean QTc at intoxication was 0.446 s (SD 0.06). The mean QTc pre-intoxication was 0.426 s (SD 0.03). The mean QTc post-intoxication was 0.420 s (SD 0.03). QTc recorded on ECG_INTOX_ was not significantly longer compared to ECG_PRE-INTOX_ or ECG_POST-INTOX_ for the whole sample (*p* = 0.104, *p* = 0.070), men (*p* = 0.367, *p* = 0.263), women (*p* = 0.136, *p* = 0.113) or acute intoxications only (*p* = 0.556, *p* = 0.859). For chronic intoxications, QTc was significantly longer in ECG_INTOX_ compared to ECG_POST-INTOX_ (*p* = 0.040), but not compared to ECG_PRE-INTOX_ (*p* = 0.145) (Figure 2).

QTc on ECG_INTOX_ correlated only weakly with s-Li at intoxication (ρ = 0.329, *p* = 0.014) (Figure 3). However, there were 13 (24%) episodes with prolonged QTc on ECG_INTOX_. Of these, nine (69%) occurred in women and eight (62%) in the context of a chronic intoxication (Figure 3). A U-wave was present in three (23%) episodes (Table 5). In seven (54%) of the 13 episodes, QTc exceeded 500 ms. Of these, five (71%) concerned chronic intoxications. The largest QTc increase was 258 ms, from 433 ms in the ECG_PRE-INTOX_ to 691 ms at intoxication. In seven of the 13 episodes, QTc reverted to normal in the ECG_POST-INTOX_. For all 13 episodes, both mean and median s-K were 3.7 mmol/L (SD 0.70, min 2.80, max 4.80 mmol/L). Seven patients had at least one known cardiovascular condition at time of lithium intoxication. None had diabetes. In six episodes, patients had been prescribed additional medicines with known or possible risk of TdP (Table 5).

On univariate analysis, only severe lithium intoxication and hypokalaemia were significantly associated with QT prolongation (Table 6).

### 3.5. T-Wave Inversion

T-wave inversion was found in 42% of episodes at the time of intoxication. There was no significant difference in the presence of T-wave inversion between ECG_INTOX_ and ECG_PRE-INTOX_ (*p* = 0.454), or ECG_POST-INTOX_ (*p* = 0.344).

## 4. Discussion

In our study, lithium intoxication led to a significant decrease in heart rate. However, the difference was small and not clinically relevant in most cases. QTc correlated only weakly with s-Li. For acute intoxications, there were no statistically significant differences in mean QTc between ECG_INTOX_ and the reference ECGs. For chronic intoxications, there was a statistically significant difference in mean QTc and ECG_POST-INTOX,_ but not ECG_PRE-INTOX_. Since our study covered a fairly small number of patients, the absence of significant mean differences could be the result of the limited sample size. 

However, about a quarter of all patients experienced QT prolongations in connection with a lithium intoxication. This was more common in chronic intoxications, but it also occurred in acute on therapeutic or acute intoxications. In approximately half of the intoxications with QT prolongation, QTc exceeded 500 ms. A QTc ≥500 ms has been identified as a powerful predictor of 30-day all cause-mortality [25]. In our study, a substantial number of patients had therefore been at risk of life-threatening complications, which could be easily missed when only exploring the summary statistics. On univariate analysis, severity of lithium intoxication and hypokalaemia were significantly associated with QT prolongation. This is in line with clinical experience. To our knowledge, this is the first study systematically exploring ECG changes during lithium intoxications and comparing these with reference ECGs before and/or after. 

The aforementioned French study, exploring ECG changes associated with lithium intoxications in an ICU setting, reported 20% collapse/shock, 2% asystole, and 2% sudden bradycardia associated with either a third-degree atrioventricular block or a sinoatrial block [12]. In this study, only 27% had chronic intoxications. Of patients with s-Li > 5.2 mmol/L, 46% were not dialyzed. Another study examined ECG changes associated with therapeutic s-Li in 53 patients four and twelve months after starting lithium. At four- and twelve-month follow-ups, significant decreases in HR and T-wave flattening were noted. Additionally, at 12-month follow-up, the PQ interval was significantly longer. However, there were no R, ST segment, or QTc changes [26]. 

Distinguishing between acute and chronic intoxications is of clinical relevance. In acute intoxications, s-Li poorly reflects intracellular lithium levels for two reasons. Firstly, lithium may not yet have been absorbed enterally. Secondly, in patients with normal renal function, during transition into the intracellular space, lithium will continuously be cleared. Chronic intoxications often occur in patients with declining renal function [7]. Here, s-Li rises more slowly in the blood. This gives lithium time to reach the intracellular space and unfold its toxic effects [22,27,28]. Patients with chronic intoxications may therefore be at particular risk. Once lithium has entered the intracellular space, it is not easily cleared [12,22,27,28]. This raises the question at which s-Li patients should be offered extracorporeal toxin removal (ECTR), such as dialysis, to clear lithium. Depending on the presence or absence of renal impairment, thresholds between 4 and 5.2 mmol/L have been recommended [12,29]. Our own work, with a higher proportion of chronic intoxications, suggests a lower threshold to decrease the risk of cardiac complication and mortality [7,22].

Being observational, our current study depended on the quality of the information recorded in the medical records. ECG was available in only about half of all episodes, not all of which had reference ECG available. The quality was generally good. We only selected episodes with s-Li ≥1.5 mmol/L to avoid a bias towards mild and borderline intoxications. This bias towards more severe intoxications ensured that we did not underestimate the impact of lithium on cardiac conduction. The clinical course of lithium intoxications is often difficult at the time of presentation. Early dialysis may prevent the development of more serious cardiac complications associated with higher lithium concentrations. In our region, the threshold for dialysis was low [7]. In our sample, dialysis was used as a treatment for intoxication in 11 episodes. The lowest s-Li at which dialysis was used was 1.9 mmol/L. Therefore, our findings should not be extrapolated to clinical settings that employ higher thresholds of s-Li for dialysis [22].

## 5. Conclusions

Our findings suggest that lithium intoxication can affect HR and QTc. On summary statistics, these changes seem mostly discrete and not clinically relevant for many patients. However, some patients develop severe QT prolongations. This can occur in all types of intoxications—chronic, acute on therapeutic, or acute. Higher s-Li and hypokalaemia may add to the risk of prolonged QT. Therefore, an ECG should always be taken with patients who present with lithium intoxication.

## Figures and Tables

**Figure 1 jcm-11-05941-f001:**
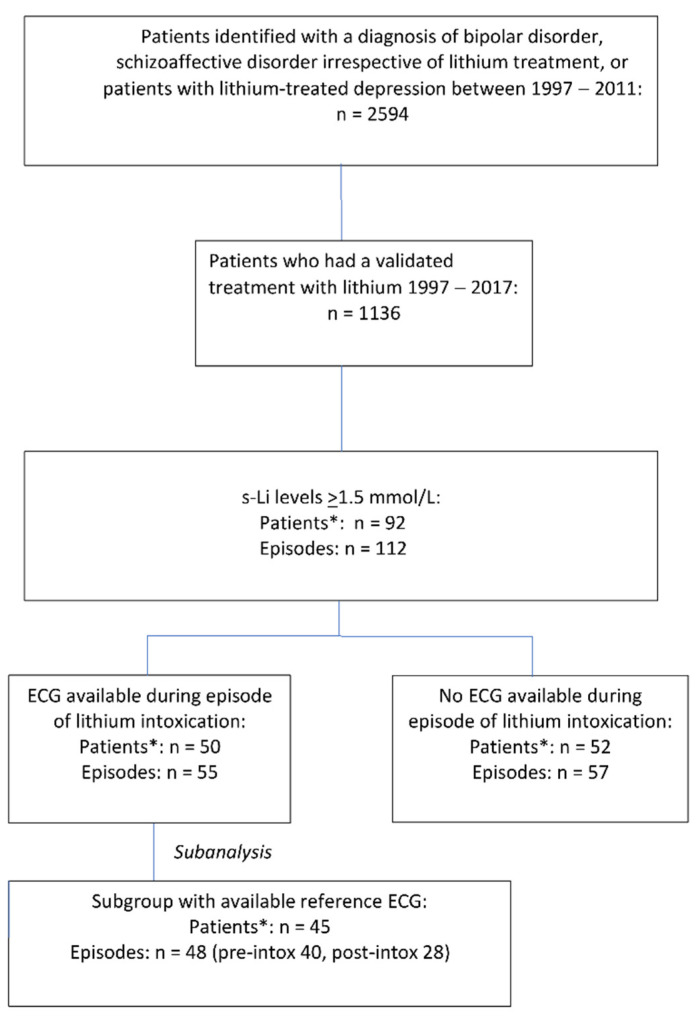
Selection of study sample. n: number. * Some patients had more than one episode of lithium intoxication.

**Figure 2 jcm-11-05941-f002:**
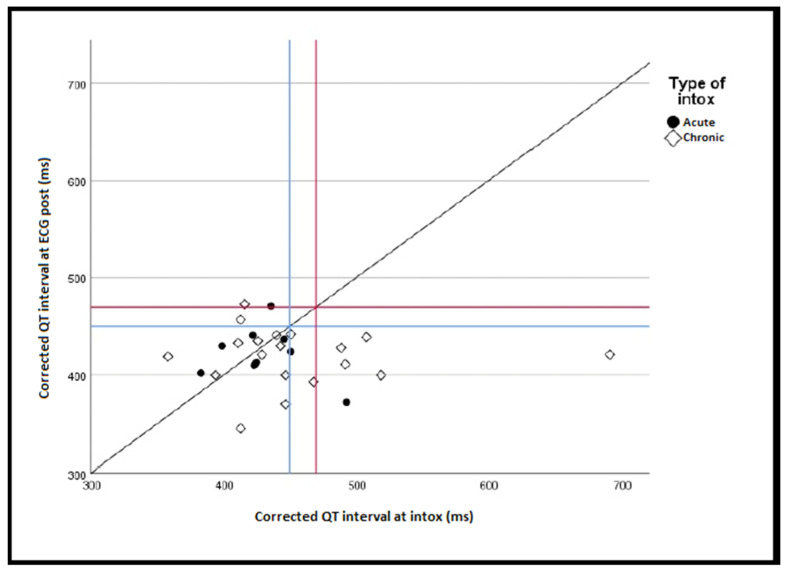
Relationship between QTc after and at the time of lithium intoxication (*n* = 28). Blue lines at 450 ms indicating cut-off point for QT prolongation among men. Red lines at 470 ms indicating cut-off point for QT prolongation among women. Intox: intoxication; ms: millisecond.

**Figure 3 jcm-11-05941-f003:**
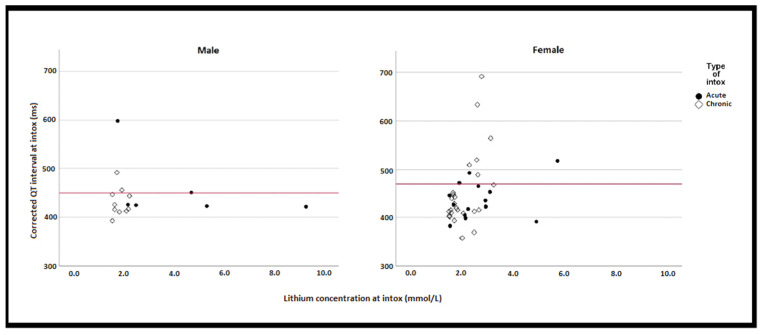
Relationship between lithium concentrations and QTc. Red lines indicating cut-off point for QT prolongation (450 ms for men and 470 ms for women). Intox: intoxication; ms: millisecond.

**Table 1 jcm-11-05941-t001:** Selection criteria for ECG.

ECG	Criteria
**ECG_INTOX_**	Acute intoxication: ECG within 24 h from detected s-Li ≥ 1.5 mmol/L Chronic intoxication: ECG within four days from detected s-Li ≥ 1.5 mmol/L
**ECG_PRE-INTOX_**	ECG taken at least one day before lithium intoxication
**ECG_POST-INTOX_**	ECG taken later than four days after lithium intoxication

ECG: electrocardiogram; s-Li: serum lithium concentration.

**Table 2 jcm-11-05941-t002:** Criteria for ECG changes.

ECG Changes	Criteria
**PQ interval**	Prolonged if >220 ms
**Heart rate**	Bradycardia < 50 b/min Tachycardia > 100 b/min
**QT prolongation**	QTc > 470 ms for females, QTc > 450 ms for males f a U-wave was present, the QT interval was measured from the beginning of the Q-wave to the nadir on the U-wave.
**ST segment elevation**	If present in two adjacent leads with ≥2 mm for males and ≥1.5 mm for females in V2-V3, or ≥1 mm in other leads.
**ST segment depression**	If present with ≥1 mm in two adjacent leads
**T-wave changes**	If inverted
**U-wave**	If present

ECG: electrocardiogram; ms: millisecond; b/min: beats per minute; QTc: corrected QT interval; mm: millimetre.

**Table 3 jcm-11-05941-t003:** Baseline characteristics of the sample at time of intoxication (n = 55 episodes).

Age, mean (SD)	55.2 (18.5) years
Age, median (min-max)	56.0 (22–86) years
Sex, n (%) *Male* *Female*	16 (29) 39 (71)
Type of intoxication, n (%) *Acute* *Chronic*	20 (36) 35 (64)
Lithium concentration at time of intoxication, mean (SD)	2.4 (1.3) mmol/L
Lithium concentration at time of intoxication, median (min-max)	2.1 (1.5–9.3) mmol/L
Severity of intoxication, n (%) *Severe* (>2.5 mmol/L) *Moderate* (≤2.5 mmol/L)	16 (29) 39 (71)
Dialysis treatment, n (%)	11 (20)
Lithium concentration in episodes treated with dialysis, median (min-max)	2.7 (1.9–9.3) mmol/L
Lithium concentration in episodes treated without dialysis, median (min-max)	1.8 (1.5–5.3) mmol/L
Time between lithium serum concentration measurement and ECG, mean (SD)	1.0 (8.0) h
Pre-existing cardiac and vascular comorbidities at time of intoxication, n (%) *Total* *Cerebral stroke/TIA* *Ischemic heart disease* *Heart failure* *Angina pectoris* *Atrial fibrillation* *Bradycardia* *Hypertension* *Pace-maker use*	28 (51) 5 (9) 3 (5) 6 (11) 2 (4) 3 (5) 1 (2) 25 (45) 1 (2)
Cardiac risk factors at time of intoxication, n (%) *Diabetes mellitus type 1* *Diabetes mellitus type 2* *Hyperlipidaemia*	3 (5) 7 (13) 8 (15)
Presence of medications with known risk of TdP, n (%) ^a^	12 (22)
Presence of antiarrhythmic medication including beta blockers, n (%) ^a^	13 (24)
Presence of antihypertensive medication excluding beta-blockers, n (%) ^a^	23 (42)
Potassium at time of intoxication, mean (SD) ^b^	4.3 (0.8) mmol/L
Potassium concentration at time of intoxication, median (min-max) ^b^	4.2 (2.8–7.3) mmol/L
Hypokalaemia ^b,c^, n (%)	8 (15)
Hyperkalaemia ^b,d^, n (%)	13 (24)

n: number; SD: standard deviation; ECG: electrocardiogram; TIA: transient ischemic attack; TdP: torsades de pointes. ^a^ episodes could have medications from more than one category. ^b^ available for 51 episodes. ^c^ <3.5 mmol/L from January 1999 until June 2004, <3.6 mmol/L since June 2004. ^d^ >4.7 mmol/L from January 1999 until June 2004, >4.6 mmol/L since June 2004.

**Table 4 jcm-11-05941-t004:** ECG changes at time of intoxication (n = 55 episodes).

Heart rate, n (%) *Bradycardia* *Tachycardia*	3 (5) 3 (5)
Rhythm, n (%) *Sinus rhythm* *One beat sinus arrest* *Nodal rhythm* *Atrial fibrillation* *Pacemaker rhythm*	48 (87) 1 (2) 2 (4) 4 (7) 1 (2)
PQ interval, n (%) *Prolongation ^a^* *AV block type 1*	2 (4) *2 (4)*
Branch block, n (%) *LBBB* *RBBB*	1 (2) 2 (4)
QT prolongation, n (%)	13 (24)
ST segment elevation, n (%) ^b^	0 (0)
ST segment depression, n (%) ^b^	2 (4)
T-wave inversion, n (%) ^c^	23 (42)
U-waves, n (%)	4 (7)

ECG: electrocardiogram; n: number; LBBB: left bundle branch block, RBBB: right bundle branch block. ^a^ available for 48 episodes with sinus rhythm. ^b^ available for 53 episodes, excluding one telemetry ECG and one ECG with LBBB. ^c^ available for 54 episodes, excluding one telemetry ECG.

**Table 5 jcm-11-05941-t005:** Episodes of lithium intoxication with QT prolongation.

Episode (n = 13)	Type of Intoxication	s-Li (mmol/L)	s-K (mmol/L)	QTc at Intoxication (s)	QTc in Reference ECG (s)	Other ECG Findings	Drugs with Known or Possible Risk of TdP	Drugs with Conditional Risk of TdP	Cardiac Comorbidity
**1**	Chronic	2.78	2.90	0.691	Pre: 0.433 Post: 0.421		Aripiprazole	Bendroflumethiazide Furosemide	Hypertension Angina pectoris
**2**	Chronic	2.62	3.70	0.633		U-wave Inverted T-wave	None	None	None
**3**	Acute on therapeutic	1.76	3.40	0.598		Tachycardia	Citalopram Haloperidol Chlorprothixene	None	None
**4**	Chronic	3.13	3.60	0.564	Pre: 0.440	RBBB Inverted T-wave	None	Omeprazole Furosemide	Hypertension
**5**	Chronic	2.59	3.70	0.519	Pre: 0.394 Post: 0.400	Bradycardia	None	Bendroflumethiazide	Hypertension
**6**	Acute on therapeutic	5.73	4.70	0.517	Pre: 0.443		Levomepromazine Promethazine	Clomipramine Loperamide	None
**7**	Chronic	2.30	3.90	0.508	Pre: 0.543 Post: 0.439	Atrial PM rhythm U-wave	Venlafaxine	Olanzapine	Heart failureSick sinus syndrome (PM)
**8**	Acute	2.30	3.10	0.493	Post: 0.372		None	None	None
**9**	Chronic	1.73	4.80	0.492	Pre: 0.461 Post: 0.411	LBBB Inverted T-wave AV block I	None	Olanzapine Omeprazole	Hypertension
**10**	Chronic	2.64	2.80	0.489	Pre: 0.400 Post: 0.428	U-wave Inverted T-wave	Citalopram Clozapine	None	None
**11**	Acute on therapeutic	1.91	4.30	0.472	Pre: 0.422	LVH	None	Hydroxyzine Formoterol	Hypertension
**12**	Chronic	1.93	3.00	0.456	Pre: 0.468		Clozapine	Loperamide	Hypertension
**13**	Acute on therapeutic	4.69	4.60	0.451	Post: 0.424	Tachycardia	None	Clomipramine	None

n: number; s-Li: serum lithium concentration; s-K: serum potassium concentration; s: second; QTc: corrected QT interval; ECG: electrocardiogram; TdP: torsades de pointes; LBBB: left bundle branch block; RBBB: right bundle branch block; AV block: atrioventricular block; LVH: left ventricle hypertrophy; PM: pacemaker.

**Table 6 jcm-11-05941-t006:** Factors associated with QT prolongation at intoxication.

	OR [95% CI]	*p*
**Lithium concentration (mmol/L)** **>2.5 mmol/L = Severe** ≤2.5 mmol/L = Moderate (baseline)	**4.28 [1.15–15.95]**	**0.030**
**Age** ≥65 years <65 (baseline)	1.25 [0.35–4.53]	0.734
**Sex** Female Male (baseline)	0.90 [0.23–3.49]	0.879
**Type of intoxication** Chronic Acute (baseline)	0.89 [0.25–3.21]	0.857
**Potassium concentration (mmol/L)** **Hypokalaemia ^a^** Hyperkalaemia ^b^ Normal (baseline)	**6.67 [1.23–36.06]** 0.73 [0.13–4.19]	**0.028** 0.722
**Co-medication with drugs with known or possible risk for QT prolongation** Yes No (baseline)	0.53 [0.15–1.85]	0.318
**Concomitant use of medicines with other cardiac effects** Yes No (baseline)	1.17 [0.34–4.06]	0.809
**Cardiovascular comorbidities and risk factors** Yes No (baseline)	1.17 [0.34–4.06]	0.809

OR: odds ratio; CI: confidence interval; ^a^ <3.5 mmol/L from January 1999 until June 2004, <3.6 mmol/L since June 2004; ^b^ >4.7 mmol/L from January 1999 until June 2004, >4.6 mmol/L since June 2004.

## Data Availability

The datasets generated and/or analysed during the current study are not publicly available due to lack of ethics committee permission and not having been part of the consent process. The structures of the dataset and the coding specification are available from the authors. Any other reasonable request will be raised with the Swedish Ethical Review Authority and the healthcare provider.

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
