# Peer review of "Effects of Toxic Lithium Levels on ECG—Findings from the LiSIE Retrospective Cohort Study"

_jcm, 2022, doi:10.3390/jcm11195941_

Round 1

Reviewer 1 Report

The manuscript "Effects of toxic lithium levels on ECG – findings from the LiSIE 2 retrospective cohort study" investigates changes in ECG during Lithium intoxication. The main finding is an increase in QTc which is only observed in chronic intoxications.

My main concern is the relevance of the reported results, since it is concluded that overall results suggest that lithium intoxication has no clinically significant effects on ECG. This conclusion might be due to the main limitation of this study which is the very limited sample size which is too small to capture discrete effects.

Are the reported results sufficient to indicate an ECG in all patients with Lithium intoxications admitted into emergency units? It seems that, from the obtained results, it might only be necessary in patients with chronic intoxications.

Reviewer 2 Report

This study appears to be well done.  That said, the only major critique I have is wondering why multivariate analyses were not conducted?  While descriptive statistics and bivariate analyses are provided, why not conduct a multivariate regression to ensure that relationships of interest are not confounded and to determine whether demographic characteristics are relevant for understanding these relationships?
